# *Serratia marcescens* Outbreak at a Correctional Facility: Environmental Sampling, Laboratory Analyses and Genomic Characterization to Assess Sources and Persistence

**DOI:** 10.3390/ijerph20176709

**Published:** 2023-09-04

**Authors:** Donna Ferguson, Rahil Ryder, Rawni Lunsford, Arie Dash, Amanda Kamali, Akiko Kimura, John Crandall, Rituparna Mukhopadhyay, Heather Dowless, Nancy Ortiz, Nathaniel K. Jue

**Affiliations:** 1Public Health Laboratory, County of Monterey Health Department, Salinas, CA 93906, USA; 2Public Health, Medical Services Division, California Correctional Health Care Services, Elk Grove, CA 95758, USA; 3Infectious Diseases Branch, Center for Infectious Diseases, California Department of Public Health, Richmond, CA 94804, USA; 4Microbial Diseases Laboratory Branch, Center for Laboratory Sciences, California Department of Public Health, Richmond, CA 94804, USA; 5Department of Biology and Chemistry, California State University, Monterey Bay, Seaside, CA 93955, USA

**Keywords:** *Serratia marcescens*, environmental microbiology, outbreak, correctional facility, virulence factors, AMR, whole genome sequencing and investigation

## Abstract

*Serratia marcescens* is an environmental bacterium and clinical pathogen that can cause an array of infections. We describe an environmental sampling and comparative genomics approach used to investigate a multi-year outbreak of *S. marcescens* at a correctional facility. Whole genome sequencing analysis revealed a predominant cluster of clonally related *S. marcescens* from nine patient cases and items associated with illicit drug use. Closely related strains found among items associated with case-patient cells and diluted Cell Block 64 (CB64), a quaternary ammonium disinfectant, and Break Out (BO), a multipurpose cleaner, highlighted their role as environmental reservoirs for *S. marcescens* in this outbreak. Comparative genomic analysis suggested outbreak strains were both persistent (identical strains found over long periods and in multiple locations of the correctional facility) and diverse (strains clustered with multiple global samples from NCBI database). No correlation was found between antimicrobial resistance (AMR) genes of outbreak strains; NCBI strains have more AMR genes. Principal component analysis (PCA) of virulence factors associated with persistence and infectivity indicated variation based on phylogroups, including the predominant cluster; identifiable variations among environmental versus clinical strains were not observed. Identification of multiple distinct genetic groups highlights the importance of putting epidemiological genomic studies in a proper genetic context.

## 1. Introduction

*Serratia marcescens*, an environmental bacterium commonly found in soil, water and plants, can cause opportunistic infections, including urinary tract infections, wound infections, bacteremia, and endocarditis. The organism has been associated with more than 100 outbreaks worldwide [1]; sources of outbreaks in nosocomial settings include contaminated medical devices, syringes, colonized patients, contaminated hands [1,2] and quaternary ammonium disinfectants [3].

Whole genome sequencing (WGS) is a powerful tool used in outbreak investigations to characterize microbial strain types from suspected sources and determine whether they are identical or closely related to strains from infected individuals. Finding sources of *S. marcescens* outbreaks may be challenging if the organisms are ubiquitous in the environment and comprise many strain types. Thus, a multimodal approach combining targeted sampling of suspected sources, epidemiological data, and knowledge about the genomic diversity of *S. marcescens* strains is critical to find matching or closely related strains among infected individuals and sources.

Moreover, genomic sequencing of genetic features such as virulence factors can further understanding of enhanced capacity for pathogenicity and persistence among outbreak strains. The presence of fimbriae, flagellar and biofilm genes associated with bacterial movement, attachment and colonization indicate enhancement of bacterial persistence in individuals and environmental reservoirs [4]. Proline metabolism can assist with stress protection and energy production [5], while hemolysin and siderophore genes allow bacteria to acquire iron from host cells for growth and progression of disease [6].

A large, prolonged outbreak of *S. marcescens* infections at a correctional facility was investigated from March 2020 through December 2022 in California. Twenty-four case-patients with *S. marcescens* serious infections were identified, including patients with epidural abscess, osteomyelitis, and endocarditis; most reported recent injection drug use (IDU). Environmental sampling and laboratory analyses were initiated to support the epidemiological investigation. This report follows the epidemiological investigation and focuses on the high-quality assembly and annotation of whole genome, short-read sequenced *S. marcescens* strains to confirm identification and assess genome diversity, and comparative analyses of antimicrobial resistance and virulence factors. The environmental sampling approach used to identify potential sources of infection and laboratory experiments to assess *S. marcescens* survival in Cell Block 64 (CB64), a quaternary ammonium disinfectant used at the correctional facility, are also described.

## 2. Materials and Methods

### 2.1. Environmental Sampling

A total of 152 environmental samples were collected from November 2020 through September 2021 and analyzed for *S. marcescens* contamination. Water samples were collected from holding tanks, wells (pre- and post-reverse osmosis treatment), a drinking water fountain and sink faucets in cells (living quarters) of incarcerated persons and a breakroom. Swabs were used to sample reverse osmosis filters, water hoses in janitor closet, floors near doorways of case-patient cells, shower heads and walls in communal showers, and other areas exposed to water and covered with pink to red pigmented biofilm or “slime” associated with *Serratia* colonization. Since correctional facility-issued disinfectants used throughout the facility were suspected as a potential source of contamination, we sampled items associated with CB64 (https://catalog.calpia.ca.gov/product/cell-block-64-disinfectant-cleaner-deodorizer-3961/, accessed on 30 June 2023) and Break Out (BO), a multipurpose cleaner and degreaser (https://catalog.calpia.ca.gov/product/break-out-multipurpose-cleaner-and-degreaser-2446/, accessed on 30 June 2023) (California Prison Industry Authority, Lancaster, CA, USA). These items included unopened bottles of concentrated CB64, surfaces of dilution machines and tubing used to store and dilute CB64 and BO with tap water, and re-used CB64 and BO bottles and trash cans and mop buckets used by incarcerated persons for mixing, diluting and storing CB64 and/or BO. Samples obtained from cells included re-used containers for storing CB64 and BO, scrubbies (scrub pads), a drinking water bottle containing tap water, a bar of soap, toothbrushes, eating and drinking utensils and a plastic bottle used as urinal. Since 22 case-patients reported IDU prior to infection, we obtained needles/syringes surrendered by incarcerated people and items used to store drugs such as bottles of nasal spray, prayer oil, Hibiclens^®^ (4% chlorhexidine gluconate) and rinsates from sharps containers at clinics at the correctional facility used for disposal of needles. Hand-rinsates from incarcerated persons were collected by rinsing of hands with sterile saline.

### 2.2. Isolation, Culturing, and Identification of S. marcescens

Water samples (100–500 mL) were filtered onto 47 mm diameter 0.45 µm pore size mixed cellulose ester membranes, placed onto chromogenic *Serratia* agar (CHROMagarTM Paris, France) and incubated at 37 °C for up to 48 h [7]. Liquids including disinfectants, hand-rinsates of incarcerated people, and rinsates from containers were tested similarly as water samples. Swabs were initially streaked directly onto *Serratia* CHROMagarTM, MacConkey (MAC) and Blood Agar (BA) plates and then incubated in Brain Heart Infusion (BHI) broth for up to 5 days; BHI broth with growth was subcultured to *Serratia* CHROMagarTM. Needles, syringes and scrubbies were placed directly into BHI broth and cultured similarly as above. Colonies presumptive for *Serratia* were identified to species level using Gram stain, Oxistrips ™ (Hardy Diagnostics, Santa Maria, CA, USA) and API 20E (bioMérieux, Inc., Durham, NC, USA). Cultures obtained from hospitalized patients were confirmed as *S. marcescens* and/or *Serratia* species as described above. Multiple *S. marcescens* isolates from the same sample were labeled “–1", "–2", etc. per isolate as part of the sample identification number. *S. marcescens* isolates were sent to California Department of Public Health (CDPH), Center for Laboratory Sciences Microbial Diseases Laboratory Branch for WGS.

### 2.3. Lab Experiment to Assess S. marcescens Survival in CB64

A known concentration (1.5 × 10^8^ colony forming units per milliliter (CFU/mL)) of *S. marcescens* from a case-patient was inoculated into CB64 (diluted and undiluted) and controls (sterilized tap water and 0.85% sterile saline), allowed to incubate at room temperature and enumerated by culture to compare changes in concentration immediately after inoculation and at various time points thereafter for up to 7 days. Inoculated solutions included CB64 that was undiluted, diluted in tap water 1:64 (as per manufacturer’s instructions for use) and diluted further (1:1000 and 1:10,000). Diluted CB64 included “unexpired” solutions (<24 h old after dilution as per manufacturer’s recommendations for use) and “expired” solutions (≥24 h old after dilution). Enumerations were performed by removing known volumes of solutions that were centrifuged (1000× *g* for 20 min) and rinsed twice in 0.85% sterile saline, resuspended in saline and cultured using BHI agar. After 24 h incubation at 35 °C, colonies on BHI agar were enumerated as CFU/mL.

### 2.4. Whole Genome Sequencing

*Serratia* isolates from environmental samples and clinical isolates were sequenced and analyzed by the CDPH Center for Laboratory Sciences Microbial Diseases Laboratory Branch using an in-house validated protocol via an Illumina MiSeq (Illumina, San Diego, CA, USA). Sequences are publicly available in the National Center for Biotechnology Information (NCBI) database (bioproject number: PRJNA981498 and NCBI Biosample numbers) (Appendix A). Briefly, genomic DNA was extracted using DNeasy Blood & Tissue Kit (Qiagen, Germantown, MD, USA). Bacterial DNA quality and quantity were checked using NanoDrop and Qubit (ThermoFisher, Waltham, MA, USA). The Illumina DNA prep library preparation procedure was used to generate the library, and whole genome sequencing was performed on an Illumina MiSeq sequencer using 2 × 250 cycle MiSeq sequencing kits. The library quality and quantity were checked using a 2100 BioAnalyzer Instrument (Agilent, Santa Clara, CA, USA) and Qubit. The Illumina PhiX sequencing control was used as internal control in the sequencing run. Genomes were generated with a depth coverage greater than or equal to 60× [8].

### 2.5. Genome Assembly, Annotation, and Relatedness

To assemble the whole genome sequences, the MiSeq reads were first trimmed using Trimmomatic (v0.39) [9], de novo assembled using Unicycler (v0.5.0) [10]. The resulting contigs were then ordered and oriented to a reference genome assembly (GCF_003031645.1) using RagTag (v2.1.0) and Minimap2 [11]. The reference genome was selected based on the highest average percent coverage (>93%) and percent mapped (>73%) from a trial of other NCBI *S. marcescens* genomes with one clinical and one environmental sequence from the outbreak. The resulting assembled genome included all scaffolds and unmapped contigs. To ensure high-quality, complete genomes, Benchmarking Universal Single Copy Orthologs (BUSCO) (v5.4.3) was used to determine completeness of the assembly [12], and Quast (v5.0.2) was used to collect the N50, length of the assembly, and GC % [13]. Prodigal (v2.6.3) was used for gene predictions [14]. These assembly and QC steps were run on Terra (https://app.terra.bio/) (accessed on 28 June 2023), a cloud-based platform that allows for easily scalable workloads and exposes computational resources to laboratories and other research entities that would otherwise not have access to them. Code (https://github.com/MontereyCoPHL/Serratia_pipeline) (accessed on 30 June 2023) and importable workflows (https://dockstore.org/workflows/github.com/MontereyCoPHL/Serratia_pipeline/Serratia_Assemble:main?tab=info) (accessed on 30 June 2023) are publicly available. To identify the genes, DIAMOND blastp (v2.0.8) was used against the nr protein database downloaded on 1 March 2022 indexed with new_taxdump, and prot.accession2taxid from 7 March 2022 was used for taxonomy identification [15]. To compare broader patterns of genome relatedness, we calculated pairwise average nucleotide identities (ANIs) for the largest scaffolds from each assembly using OrthoANI (v0.90) [16], thus eschewing possible plasmid-related sequences in this analysis.

### 2.6. Antimicrobial Resistance (AMR) and Virulence Factors

The NCBI AMRfinder (v3.11.4) was used to identify antibiotic resistance genes for all sequences including background sequences from NCBI [17]. A subset of 27 sequences were used for the virulence factor analysis. A manual search for virulent genes was conducted using the annotations from the DIAMOND blastp results. Several *S. marcescens* virulence factor genes are listed in the Abreo and Altier study [18]. A key word search for all virulence gene names was conducted using “virulen” (virulence), “virb”, “hemolysin”, “hly”, “siderophore”, “flagel” (flagellar/flagellum), “biofilm”, “fimbria”, “proline”, “betain”, “toxin”, and “lipopolysaccharide”. For both AMR and virulence factor analysis, all contigs in the genome were used, including those potentially associated with plasmids.

### 2.7. Principal Component Analysis (PCA) of Virulence Factors

PCA was conducted using the program PAST (v4.0.3) [19]. The virulence factors were grouped. Proteins with “virulence” and “virb” were classified under virulence regulation. “Hemolysin” and “hly” were classified under hemolysin. “Siderophore” had its own classification to notate differences with hemolysin. “Flagellar” and “biofilm” were the only key words for flagellar and biofilm proteins, respectively. “Fimbria” and “fimc” were classified under fimbrial. “Proline” and “betain” were classified under Proline/betaine. “Toxin” and “lipopolysaccharide” were classified under toxin. Standardization was performed on the virulence factor groups and the AMR counts.

### 2.8. Construction of Trees

To generate a fine-scaled perspective on genetic relatedness, we created detailed analyses of genetic similarity using both phylogenetic methods and specific single nucleotide polymorphism (SNP) variant calls. The phylogenetic tree was created with Realphy (v1.13) [20] using standard settings and the reference sequence GCF_003031645.1, which was used as the reference to assemble the genomes. *S. marcescens* sequences were selected from Abreo and Altier [18], downloaded from NCBI for global context, and an *S. liquefaciens* sequence was used to root the tree. Archaeopteryx 0.9928 beta was used to visualize and edit the phylogenetic tree [21]. Bars on the right of the tree were manually added to display phylogeny group, sample type (clinical vs. environmental), and AMR with one visualization. Phylogroups were assigned based on grouping into a polytomy on the phylogenetic tree, >99% ANI, and <1000 SNP difference. Phylogroups represent highly, genetically similar *S. marcescens* variants and could consist of strains coming from either patients or environmental sampling. When we refer to a “cluster” in an epidemiological context, this would be a phylogroup that possesses essentially the same genomic variant in multiple patients.

To generate the SNP-based tree, paired-end reads were quality trimmed at the threshold of Q30 and were mapped to a different reference genome than the above phylogenetic tree (NZ_CP020503) using QIAGEN CLC Genomics Workbench 10.1.1. The reference genome used for SNP-based phylogenetic analysis was selected using the KmerFinder 3.2 webtool by Center for Genomic Epidemiology (CGE, https://www.genomicepidemiology.org/, accessed on 30 June 2023). The selected reference genome should be a complete genome and closely related to the strains for the SNP-based analysis. The repeat regions of the reference genome were masked using the PHAST webtool (http://phast.wishartlab.com/, accessed on 30 June 2023) and not included in the analysis. After mapping, the BAM files generated were processed using a customized shell script (https://github.com/ritumukh/MiSeq_CDPH, accessed on 30 June 2023) to automate the subsequent steps that included: (1) SNP calling in both coding and non-coding regions using SAMtools mpileup (v.1.2); (2) converting into VCF matrix using bcftools (v0.1.19); (http://samtools.github.io/bcftools/, accessed on 30 June 2023); (3) variant parsing using vcftools (v.0.1.12b) to include only high-quality SNPs (hqSNPs) with ≥30× genome coverage plus quality score > 200, excluding any DNA insertion and deletion (InDels) or heterozygote call; and (4) converting the SNP matrix into a FASTA alignment file. This FASTA alignment file was imported into CLC Workbench to create a phylogenetic tree using Maximum Likelihood algorithm and 100 bootstrap replicates.

### 2.9. Epidemiological Investigation

The epidemiological investigation was conducted by investigators from the California Department of Public Health, U.S. Centers for Disease Control and Prevention, and the California Correctional Health Care Services. Investigators identified case-patients, reviewed medical histories, identified potential sources of infection, conducted multiple interviews with incarcerated persons to determine social connections and activities associated with IDU and disinfectants, assisted with environmental sampling and implemented remediation measures to control further infections.

## 3. Results

### 3.1. Detection of S. marcescens among Environmental Samples

Of 152 environmental samples analyzed, 27 (18%) were positive for *S. marcescens*, including a used needle/syringe surrendered by a case-patient, used nasal spray bottle previously containing methamphetamine, hand-rinsate from an incarcerated person, pooled water in a dilution machine located near cells of case-patients, mop bucket, trash cans and other containers used for storing, diluting and dispensing CB64 and BO, water and laundry detergent stored in a used body wash bottle, cleaner stored in a hand sanitizer bottle, coffee in a plastic cup, urinal bottle, floor surfaces of communal showers, doorways to cells and scrubby cleaning pads. *S. marcescens* was not detected in water samples representative of the facility’s water system, rinsates of used needles in sharps containers at the correctional facility clinic, swab samples of cell walls, and CB64 in original, unopened containers from the manufacturer.

### 3.2. S. marcescens Survival in CB64

*S. marcescens* inoculated into CB64 diluted at 1:64 (solutions 2–5) was nonviable immediately after inoculation (time zero) and up to 7 days later (Figure 1, Table 1). In contrast, the bacteria remained viable in CB64 diluted ≥ 1:1000 (solutions 6–9) after an initial reduction in concentrations, due to stress and/or limited biocidal effect of diluted CB64, and then increased in concentrations (~2 logs) within 48 h of inoculation, indicating bacterial persistence and growth. Interestingly, *S. marcescens* persisted in control solutions for up to 7 days, and the log reduction in tap water at 7 days was only 0.5 log lower than that of 0.85% saline.

### 3.3. Genome Assembly and Relatedness

High-quality genomes were assembled from whole genome sequencing of 60 *S. marcescens* strains from the outbreak (Appendix A). The Unicycler assembler with the reference-based RagTag scaffolding created the final corrected genome assembly, which had both high N50 values (Avg = 5,213,316) and BUSCO completeness percentages (Avg = 96.3). The average N per 100 kb was 94. The genomes for all strains contained a GC percentage average of 59.3% and an average genome length of 5,306,590 bps.

We assigned *S. marcescens* strains to 11 different phylogroups for illustrative and comparative purposes (Figure 2). There was one principal cluster of genetically similar *S. marcescens* strains (ANI ~100%/0–25 SNPs) (Appendix A) from clinical specimens from nine case-patients, a used needle/syringe and nasal spray bottle containing methamphetamine (Cluster Group 1, Figure 3). Strains in Cluster Group/phylogroup 1 demonstrated persistence over time, with samples from the first patient infected by this strain obtained on 2 January 2021 and the most recently identified patient in this cluster obtained on 13 October 2022. Phylogroups 2–11 did not represent clusters of epidemiological significance or distinct clades but did exhibit genetic similarity high enough to also warrant further investigation.

There were strains in other phylogroups (3, 4, 8 and 10) that also showed high similarities between subgroups that contained strains from clinical and environmental samples. The levels of ANI/SNP differences between these clinical and environmental strains were ~100%/39 SNPs for phylogroup 4 (Patient 14 and Sample M), ~100%/14–15 SNPs for phylogroup 8 (Patient 5, Sample B, Sample H and Sample Y), and ~100%/34 SNPs for phylogroup 10 (Patient 6 and Sample Z) (Appendix A).

Multiple *S. marcescens* isolates sampled from the same source were sequenced to assess strain diversity in the context of sampling approach. Many samples yielded sequences with phylogroups that were highly similar (ANIs = ~100%) and easily identifiable as members of the same strain of *S. marcescens*, while not being precisely identical. For instance, two isolates from scrubbies (Sample E) differed by just 16 SNPs (Appendix A), and both clustered with a strain from a surrendered used needle (Sample I) collected on the same day from the same cell (phylogroup 11, Figure 2), and four strains taken from the same needle in phylogroup 1 all differed by just 1 or 2 SNPs. A few samples, however, showed diversity within sampling targets among isolates, revealing more than one strain type (Figure 2). In phylogroup 9, there were multiple strains isolated from blood (1), urine (1) and wound (2) from the same individual on two separate dates (4 days apart). ANIs indicated that they were ~100% similar (Appendix A), and the phylogeny showed them radiating from a polytomy on the tree (Figure 2). However, SNP analysis indicated some SNP differences between these strains (10–32 SNPs), with the two strains taken from a wound on the same day differing by 22 SNPs.

Looking at a subsample of data (including 16 genomes isolated from patients and 32 from environmental sampling) for which we had location metadata, we could see that most cases were concentrated in Area IV with patients from phylogroups 1 and 10 (genetically distinct groups) (Figure 4). There were also connections between strains in phylogroups that were geographically distant from each other (phylogroups 2, 4, 6, 7, and 8). Many of these groups consisted of strains from environmental samples only, while phylogroups 4 and 8 included strains from case-patients. In addition to a wide spatial distribution of strains within the correctional facility, many strains showed extensive temporal persistence. For example, strains in phylogroup 1 spanned a range of dates from January 2021 to October 2022 and exhibited minimal genetic differences across sampling dates (Appendix A); the first strain (clinical) and last strain (clinical) obtained from a different patient from phylogroup 1 had only one SNP difference between genomes.

### 3.4. Antimicrobial Resistance and Virulence Factors

Prodigal gene prediction software predicted between 4586 and 5459 genes for the assembled genomes. Of these genomes, 27 bacterial strains representative of the variation among clusters were selected for the PCA of virulence-related gene content (bolded in Appendix A). Presence/absence of 191 virulence factor genes was used to cluster strains. Virulence factor counts ranged from 150 to 191 genes in the outbreak strains (Appendix A). Overall, approximately 60% of the total variation was explained by the first two principal components (PCs) (Figure 5; Appendix A). Biplots of those PCs showed some distinct clustering of strains, particularly phylogroup 1. Genes related to toxins, hemolysin, and virulence regulation had the three highest loading values for PC1 (0.477, 0.419, and 0.414, respectively), which explained 43.111% of the variation, while flagellar, proline/betaine and biofilm had the three highest loading values for PC2 (0.513, 0.485, and −0.434, respectively), which explained 16.048% of the variation.

While present in all strains, there was no apparent correlation of AMR genes between either clinical versus environmental strains or phylogeny groups. AMR genes identified by the NCBI AMRfinder tool showed most outbreak strains had the same three or four AMR genes (Figure 2; Appendix A), except for Sample D (scrubbie), which had eight AMR genes, including those for quinolone and streptothricin. All beta-lactamase genes in the outbreak strains were Class C.

### 3.5. Epidemiologic Investigation

During the period from March 2020 to October 2022, 24 incarcerated persons developed serious *S. marcescens* infections. Most (92%) had a history of IDU. Interviews indicated complex social connections between several case-patients, including individuals who had shared needles for IDU and tattooing or had occupied the same cells as other cases. Several case-patients reported storing diluted CB64 in their cells, which was sometimes used to clean their needles and syringes between use.

## 4. Discussion

This report describes the laboratory investigation of the first and largest known outbreak of *S. marcescens* occurring in a correctional facility in California. The investigation highlights the challenges in identifying sources of infections given the common occurrence of this organism in the environment. Environmental sampling was largely guided by epidemiological data, focusing on potential reservoirs of *S. marcescens*, case-patient residential areas (cells and communal areas), social connections and practices associated with IDU.

Initially, tap water was suspected as a source given the widespread occurrence of infected cases at the correctional facility; however, *S. marcescens* was not found in any water samples despite testing high volumes (500 mL/sample) and repeated sampling. This finding was consistent with the correctional facility’s historical water quality monitoring results; potable water samples tested monthly one year before and during the outbreak were negative for total coliform bacteria, including *Serratia* species. There were no environmental events or changes in water treatment at the onsite facility that might have affected water quality.

Since contaminated disinfectants were suspected as potential sources of *S. marcescens*, sampling then progressed to analyzing dilution machines at different housing units at the correctional facility used to dilute concentrated CB64 and BO with tap water prior to use. The investigation revealed several dilution machines had not been properly maintained and/or calibrated. Solutions of CB64 and BO sampled from different machines varied in color, indicating inconsistencies in dilution. Additionally, incarcerated people employed alternate methods for diluting and mixing CB64 solutions, including using uncleaned mop buckets and trash cans. Diluted solutions were then scooped into re-used food and drink containers for extended use in cells, not adhering to manufacturer’s instructions for diluting CB64 at 1:64 and using solutions within 24 h. Not surprisingly, *S. marcescens* was detected in items containing improperly diluted CB64 solutions. A laboratory experiment conducted by the Monterey County Public Health Laboratory using the *S. marcescens* Cluster 1 strain (Figure 2) showed survival of this organism in improperly diluted CB64 for at least 7 days. These findings are concerning, since contaminated disinfectants containing quaternary ammonium compounds such as benzalkonium chloride and 2% chlorhexidine can serve as potential sources of infection [3,22], and improper use of disinfectants can contribute to outbreaks [23].

Whole genome sequencing supported epidemiological data demonstrating that strains from nine infected patients, a needle/syringe and methamphetamine were grouped in Cluster 1, the predominant outbreak strain. Finding *S. marcescens* in scrub pads was consistent with reports that they were used by incarcerated persons to clean their cells, and at times IDU paraphernalia. Detection of *S. marcescens* in hand-rinsate from an incarcerated person suggests potential colonization of *S. marcescens* in skin and underscores the need for early intervention among incarcerated persons with wound abscesses and a history of IDU. The occurrence of *S. marcescens* contamination among these and other various items in the correctional facility emphasizes the importance of handwashing to reduce further spread of this opportunistic agent.

In this study, we sought to understand the persistence of the predominant clinical strain (Cluster 1) given the occurrence of highly diversified strains in the environment. Previous studies showed the relevance of factors that potentially enhance survival and virulence, such as AMRs, lipopolysaccharide, hemolysin, siderophore, flagellar, and biofilm in *S. marcescens* [24,25,26,27]. Franczek et al. [28] found that mannose-resistant type-K hemagglutination virulence factor was statistically higher in clinical samples out of a total of 147 *S. marcescens* strains. Piccirilli et al. [29] reported that both environmental and clinical sequences of *S. marcescens* from a neonatal outbreak had identical antibiotic resistance genes and virulence factors. As in this study, a comparison of 49 *S. marcescens* strains found on GenBank by Abreo and Altier [18] found that differences related to virulence factors were apparently more influenced by phylogeny rather than source (environmental versus clinical) of strains.

We conducted comparative analyses of high-quality whole genome assemblies to assess differences in phylogeny, virulence factors and AMR of clinical and environmental *S. marcescens* strains from the correctional facility outbreak to characterize potential differences among samples. Phylogenetic analysis suggested that *S. marcescens* outbreak strains were both persistent and diverse (Figure 2). Phylogeny groups 1, 4 and 8 included both environmental and clinical sequences with collection date differences ranging from 8 to 22 months. This long duration suggests the bacteria were persisting in the correctional facility environment, likely enhanced by adherence-related virulence factors (biofilm, flagellar, and fimbriae). Strains within phylogroup 1 (0–25 SNP differences) provide suggestive evidence for a cluster of *S. marcescens* infections associated with environmental strains from drug paraphernalia. This cluster included multiple colonies isolated from a swab of a needle used for IDU, including four genome sequences that differed by 0–2 SNPs. While suggestive of a possible mechanism of infection, at minimum, this suggests a connection between the etiological agent of infection and the environment of case-patients. Strains in phylogroup 8 were obtained from an incarcerated person and two environmental sources, all from different locations in the correctional facility, which suggests sources other than the methamphetamine and/or syringe/needle. The high diversity between this strain and that of phylogroup 1 indicated by the intervening NCBI sequences from Romania, Germany, Mexico and Japan is evidence of the diversity of isolates and potentially multiple sources of infection throughout the facility.

Patterns of genetic diversity among and within other phylogroups indicated genetic connections to different strains within the correctional facility environment. Three other phylogroups (4, 8 and 10) showed patient strains that clustered with distinct strains isolated from environmental samples. It should be noted that some of these phylogenetically defined groups demonstrated very high genomic similarities (e.g., ~100 % with ANI). Interestingly, SNP level differences were identified within most of these phylogroups, and one could make an argument that these genetic differences indicate separate strains. However, the distinct patterns of strong clustering relationships between strains by both SNP and phylogenetic analyses indicate that the genetic relationships within these clusters of samples are closer than those from more genetically distinct strains (e.g., strains from NCBI are spread throughout the tree and from very disparate sampling locations and times). Viewed within this broader context of genetic diversity, the genetic connection of these four case-patients with an environmentally isolated strain in each phylogroup suggests multiple sources of infection.

This raises the question of what SNP variation at a genome level is sufficient to constitute delineation of samples as different infective strains? There are no hard universal thresholds here. Overall, the SNP differences between strains in phylogroups 4, 8, 10 and 11 were often on the scale of 10s of SNPs, but levels of differentiation among strains, subspecies and species can vary depending on the taxonomic group [30]. Repeated sampling of the same source can provide some insight into this question, as expectations are that the genome sequences should be near identical between these samples. However, what is our definition of “nearly identical” for assembled genomes of > 5 Mb in length? In our study, specimens from the same individual (Patient 7, phylogroup 9) obtained from a wound on the same day yielded two genome sequences that differed by 22 SNPs; samples from urine and blood obtained 4 days prior differed by 10 SNPs, which differed from the wound genome sequences by 20 to 32 SNPs. Given a genome size of ~5.31 Mb and assuming observed SNP differences could be due to errors induced by either sequencing or assembly, the estimated error rate in the genome sequence would range from 0.00011% to 0.0016%. Thus, while Patient 7 could have been infected by multiple closely related strains, it could be just as likely, if not more so, that either the original strain evolved over time during infection or that this level of SNP difference is within the level of error expected in this type of genomic investigation. In other words, this level of variation among closely related isolates appears negligible. This study demonstrates that defining differences such as these as significant at a genome scale should be considered carefully and in an experimental context. The genetic distances between and among phylogroups and the inclusion of strains from NCBI aided in our analysis and improved our ability to put genetic differences between strains into context. Given this evidence from resampling and the level of differences in polytomies for our phylogroups, we would contend that strains with SNP differences on the scales of 10s, and possibly 100s, came from the same original organism, bolstering our case for strains from multiple patients being strongly genetically associated with strains obtained from environmental samples.

The spatial spread of these samples across the correctional facility suggests some level of environmental connectivity despite their physical separation (Figure 4). While the largest number of clinical strains associated with case-patients were concentrated in Area IV, genetically similar strains of *S. marcescens* were detected across locations that should be physically limited in their connectivity within the correctional facility. All phylogroups, including those with clinical strains, showed spatial spread across multiple locations in the correctional facility, even with our limited environmental sampling. Thus, mechanisms for spreading *S. marcescens* through the correctional facility environment must exist. Whether this is due to normal environmental factors, the movement of materials among areas and cell blocks associated with drug trafficking or disinfectant/cleaning solutions, or interactions among incarcerated people in shared areas, this study cannot definitively say. The data suggest that activities involving Areas IV and V could contribute to movement among “zones” and the spread of *S. marcescens* strains across the facility; however, more sampling would need to be conducted to resolve this question.

Examination of virulence factors showed that differences were most influenced by phylogroup lineage. This finding is consistent with both the Piccirilli et al. [29] and Abreo and Altier [18] studies, wherein PCA data showed no relationship found between virulence factors of clinical versus environmental samples. Instead, virulence factors were typically more similar within the same cluster on the phylogenetic tree than between them. In particular, variation in PC1, which accounted for the clustering of phylogroup 1 strains away from other clinical strains, was most strongly influenced by the number of virulence regulation, toxin and hemolysin gene counts. Additionally, strains in phylogeny groups 1–3, 5, and 6 had more virulence regulation and toxin virulence factors, while phylogeny groups 7–11 had more fimbria (Appendix A). Only one mannose-resistance fimbriae gene, described as a virulence factor in an earlier study by Franczek et al. [28], was identified in the clinical strain from Patient 7. No apparent correlation was found for AMR genes between phylogeny group or environmental versus clinical strains. In fact, the NCBI downloaded sequences used in the analyses had the most AMR genes apart from Sample D (scrubbie) from phylogeny group 5. It is common to find beta-lactamases among sequences associated with clinical specimens [18,31,32]. In this outbreak, beta-lactamase genes were found in all environmental and clinical sequences (Appendix A). Interestingly, beta-lactamases identified in this study were of Class C, extended-spectrum AmpC type, only. Overall, these results indicate that there were no clear differences in the virulence genes detected from clinical and environmental strains isolated from this investigation; however, the presence of some of these genes provides important insights into the biology of the etiological agent in this outbreak.

Our study demonstrates the utility of using a comprehensive genomic investigation of an outbreak such as this involving multiple potential environmental sources over a protracted period of time. The additional analyses performed as part of this investigation, such as the survival study defining phenotypic characteristics, comparing ANIs, establishing phylogroups using a phylogenetic tree algorithm, utilizing a broad background of NCBI genomes to provide context for genetic results, assessing the genic content (presence of AMR and other known virulence genes) of genomes, and placing genomic relationships in a spatially explicit framework, all provided us with important perspectives as to the nature of this outbreak. Using these results, we were able to identify key aspects of this outbreak, such as the multiple lines of infection, the identification of phylogroups, connections with important possible sources of infectious environmental bacteria, and possible mechanisms for the spread. In total, this work demonstrates the value of genomic data in aiding complex outbreak investigations. Despite intense and targeted but not necessarily comprehensive sampling, we were still able to find direct genetic connections between patients and contaminated environmental sources. All of this speaks to both the broad and specific value of applying genomic perspectives to this type of outbreak.

There were several limitations to this study. Isolates from all hospitalized case-patients were not available for sequencing. Limited resources restrained sampling and testing; thus, sampling was focused on areas where recent cases were identified. For most samples, multiple isolates of *S. marcescens* were subcultured for sequencing; however, only one isolate per sample was sequenced. Recovering multiple strain types from a single sample and single case-patient reflected the ubiquity of diverse strains and highlighted the challenges to finding sources for outbreaks associated with environmental bacteria. Despite using a shotgun and initial de novo assembly approach, not including long-read sequencing and DNA isolation protocols that would better target plasmid DNA could have led to an incomplete investigation of plasmid content. This, of course, could lead to issues with AMR and virulence gene detection. Overall, increasing the sample number, environmental sources, sequencing data types, and sequencing of multiple isolates (10 to 20) from samples would have allowed for clearer investigation of the relationships among strains, their plasmid content, and the connections of clinical strains with the correctional facility environment. Additionally, our ability to identify disease-associated genetic factors was also hindered by a small sample size.

The computational limitations in this study underline the need for bioinformatic resources for public health laboratories. Although cloud-based resources allowed for the incorporation of additional samples into the study, not all steps were run on the cloud due to the timeline of its availability. The computational demand and storage of sequence annotations on local resources, therefore, meant only a subset of sequences was used for the PCA virulence factor analysis. Additionally, a preferred method for the virulence factor identification would have been to use Gene Ontology (GO) terms, but due to limited local resources and knowledge of *S. marcescens*, tools to identify virulence factor GO terms could not be identified.

To further the understanding of clinical versus environmental *S. marcescens* strains, future work could implement a more robust sampling design to allow for genome-wide association studies, including a larger sample of the NCBI reference sequence database. The addition of long-read sequencing could resolve any biases created through scaffolding to a single reference genome and facilitate the clearer identification of plasmid sequences and associated gene content. Generating core genome Multi-Locus Sequence Types (cgMLST) would offer alternative methods of verifying our phylogenetic results and assist with identifying and tracking persistent strains that may be of increasingly broad public health interest. Although all genes, including those outside of the largest scaffold, were used for the AMR and virulence factor analyses, a sequencing and library construction approach that specifically targets plasmids may identify more virulence factors and AMR genes and provide insight into potential horizontal gene transfer. Finally, the inclusion of computationally intense, broad-scale genome alignments could provide us with insight into the contributions of chromosomal changes to patterns of infection. PCA analyses using all sequences from this study, including those from NCBI, would expand analyses of virulence factor distribution. Further investigation into virulence factors may enhance developments for treating infections and preventing outbreaks caused by organisms commonly found in the environment.

## 5. Conclusions

This laboratory investigation helped clarify multiple sources of infection and modes of transmission related to widespread occurrence of *S. marcescens* in the environment, including improper dilution and use of CB64 disinfectant and contaminated needles/syringes used for IDU. Genomic sequencing revealed a predominant cluster of *S. marcescens* from infected case-patients, contaminated needles/syringes and a nasal spray bottle used to store drugs in which contamination persisted for a lengthy period between cases. We also discovered examples of genetic groups that included essentially identical strain sequences that differed from the aforementioned cluster and showed strong associations between case patients and environmental samples (e.g., strains isolated from a case-patient, hand-rinsate from a patient’s cellmate and other environmental samples were also closely related).

Genomic investigations utilized in this study provided important insights into the patterns of infection, the mechanisms contributing to the spread of disease and the biological nature of the responsible etiological agent. Phylogenetic analysis suggested that *S. marcescens* outbreak strains were both persistent and diverse. The heterogeneity of strains found in clinical and environmental samples highlights the need to sequence multiple isolates from individual samples to assess strain relatedness across samples. Virulence factors related to toxins, virulence regulation, and hemolysin genes may have contributed to long-term persistence of the predominant outbreak strain. Beta-lactamases were pervasive among all strains analyzed, and no correlation was found among outbreak strains. This study underlines the importance of placing genomic data in the proper context of the epidemiological situation and the broader patterns of genetic variation in the environment.

Survival of *S. marcescens* in improperly diluted CB64 disinfectant underscores the importance of using disinfectants according to manufacturer’s recommendations and conducting routine calibration and maintenance of dilution machines that are used to dilute and dispense disinfectants.

## Figures and Tables

**Figure 1 ijerph-20-06709-f001:**
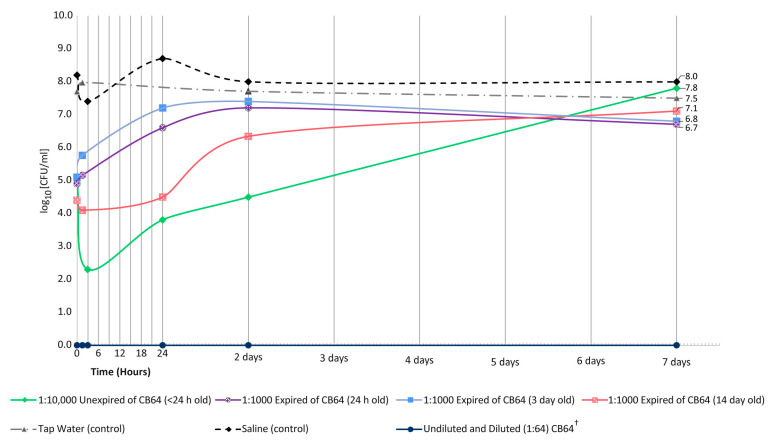
*S. marcescens* resistance in CB64 cleaning solution. Cluster 1 strain shown in Figure 2 was used as inoculum. † No growth detected for undiluted CB64 or unexpired and expired 1:64 CB64 solutions.

**Figure 2 ijerph-20-06709-f002:**
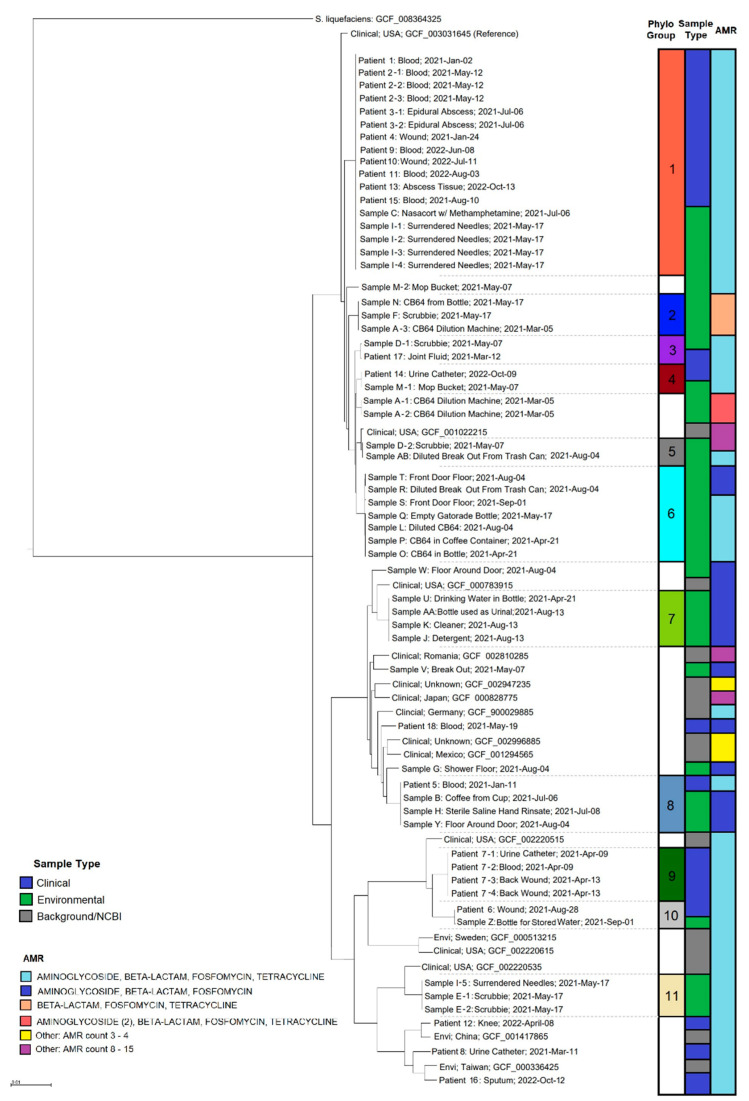
Phylogenetic tree rooted with *Serratia liquefaciens*. Nodes from the outbreak include sample identification (ID): sample source; date of collection. The left column identifies groups by phylogeny and genetic similarities, the middle column depicts sample type, and the right column depicts AMR genes. Phylogeny groups with no color indicate samples that did not group with other sequences from the outbreak. Clinical samples are blue with an ID beginning in “Patient number”, environmental samples are green with an ID beginning in “Sample Letter”. NCBI sequences are gray and labeled as: sample type; country; identifier. The reference sequence was used for the genome assembly. Scale bar is number of substitutions/site.

**Figure 3 ijerph-20-06709-f003:**
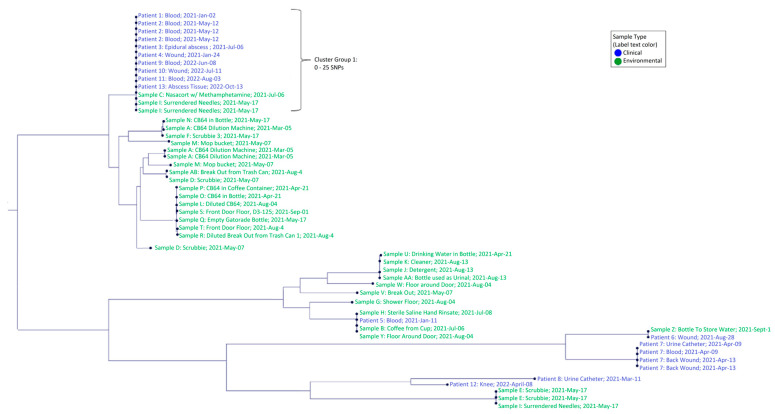
SNP-based tree of outbreak strains identifying patterns of genetic relatedness between clinical and environmental samples. Indicated in the figure is a group of strains proposed as an epidemiological cluster. Note that there are other patient strains that show high levels of genetic identity with other environmental samples.

**Figure 4 ijerph-20-06709-f004:**
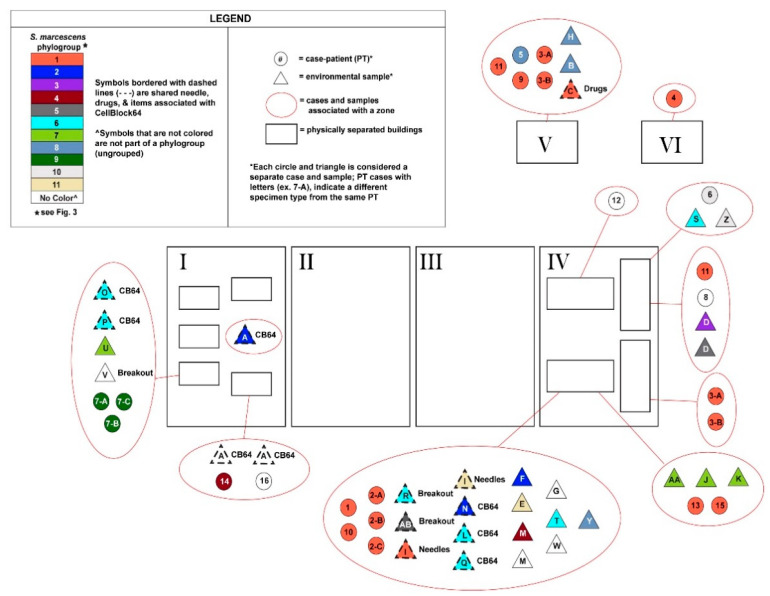
Schematic of locations of case-patients (circles)/environmental samples (triangles) at the correctional facility. Color indicates the phylogroup. Lack of color indicates no membership in a defined phylogroup. All rectangles indicate physically separated buildings or spaces. Sample descriptions are available in Appendix A.

**Figure 5 ijerph-20-06709-f005:**
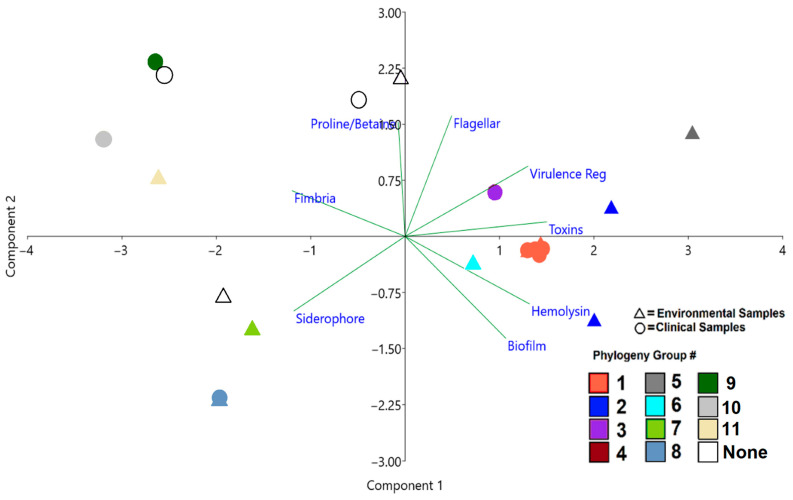
Biplot of principal components 1 and 2 from a PCA of standardized virulence factor gene counts. Samples are colorized by phylogeny group number. The triangles and circles represent environmental and clinical samples, respectively.

**Table 1 ijerph-20-06709-t001:** Survival of *S. marcescens* in CB64 and controls.

Soln. No.	Soln. Description	Ave. Concentrations—Log10 CFU/mL					
	Hours:	0	1.5	3	24	48	168 (7 days)
1	Undiluted CB64	0.0	ND	0.0	0.0	0.0	0.0
2	1:64 Unexpired CB64 (<24 h old)	0.0	ND	0.0	0.0	0.0	0.0
3	1:64 Expired CB 64 (24 h old)	0.0	0.0	ND	0.0	0.0	0.0
4	1:64 Expired CB64 (3 day old)	0.0	0.0	ND	0.0	0.0	0.0
5	1:64 Expired CB64 (14 day old)	0.0	0.0	ND	0.0	0.0	0.0
6	1:1000 Expired CB64 (24 h old)	4.9 (2) *	5.2 (3)	ND	6.6 (9)	7.2 (3)	6.7 (6)
7	1:1000 Expired CB64 (3 day old)	5.1 (0)	5.8 (13)	ND	7.2 (7)	7.4 (7)	6.8 (20)
8	1:1000 Expired CB64 (14 day old)	4.4 (15)	4.1 (14)	ND	4.5 (23)	6.3 (5)	7.1 (4)
9	1:10,000 Unexpired CB64 (<24 h old)	5.0 (6)	ND	2.3 (ND)	3.8 (0)	4.5 (14)	7.8 (18)
10	0.85% Saline (control)	8.2 (ND)	ND	7.4 (ND)	8.7 (15)	8.0 (10)	8.0 (5)
11	Tap Water (control)	7.7 (14)	8.0 (5)	ND	ND	7.7 (18)	7.5 (10)

Soln.: solution; * RSD: relative standard deviation; ND: not done.

## Data Availability

Not applicable.

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
