# Peer review of "Serratia marcescens* Outbreak at a Correctional Facility: Environmental Sampling, Laboratory Analyses and Genomic Characterization to Assess Sources and Persistence"

_ijerph, 2023, doi:10.3390/ijerph20176709_

Round 1
Reviewer 1 Report
Article titled as "Serratia marcescens Outbreak at a Correctional Facility: Environmental Sampling, Laboratory Analyses and Genomic Characterization to Assess Sources and Persistence"
There are few scientific writing issues:
The PCA in the abstract section has been used for the first time. It should be written in full format.
lines 228-250: please write all "S. marcescens" in italic format.
The discussion part is too long in my opinion.
Good job
No comments
Author Response
Point 1: The PCA in the abstract section has been used for the first time. It should be written in full format.
Response: Thank you for catching this! Corrected to “Principal Component Analysis”
Point 2: lines 228-250: please write all "S. marcescens" in italic format.
Response: S. marcescens was italicized in our original submission. For some reason this was not preserved. The formatting for data shown in Table 1 was also not preserved. We’re hoping the formatting will be corrected by the publisher.
Point 3: The discussion part is too long in my opinion.
Response: One of our authors shared this concern as well; however, all other authors wish to keep the discussion section as is.
Point 4: Good job
Response: Thank you! Your compliment is very much appreciated!
Reviewer 2 Report
This is a well written investigation of genomic/genetic analyses conducted with isolates of Serratia marcescens obtained from a correctional facility in California over a time period of two years. The authors clearly point to improper use of quaternary ammonia disinfectant as a source for spread and persistence of S.marcescens within this facility. This notion is supported by the fact that an isolate could grow in a 1:1000 dilution of CB64.
Genome sequencing of 60 isolates by Illumina MySeq and reconstructing/assembling genomes using a reference genome resulted in as the authors stated "high quality genomes" with a high completeness.
Genome comparisons led to the identification of several phylogenomic clusters revealing one cluster that was presumably mainly responsible for the outbreak, however multiple sources of infection were not excluded. PCA analyses did not reveal correlations.
As the authors discuss, the study is limited by the study design/methodology which did not allow to reconstruct the complete chromosomal/plasmidic content of the isolated strains. This is a severe limitation since genes for ABR or virulence very often reside on plasmids that can be transferred by conjugation. Additionally sequencing producing long reads (such as Oxford Nanopore) would allow de novo assembly of genomes including assigning chromosomal and plasmidic origins of genes contributing to virulence and/or antibiotic resistance. This should be clearly stated both in the introduction and discussion sections.
Author Response
Point 1: This is a well written investigation of genomic/genetic analyses conducted with isolates of Serratia marcescens obtained from a correctional facility in California over a time period of two years. The authors clearly point to improper use of quaternary ammonia disinfectant as a source for spread and persistence of S.marcescens within this facility. This notion is supported by the fact that an isolate could grow in a 1:1000 dilution of CB64.
Genome sequencing of 60 isolates by Illumina MySeq and reconstructing/assembling genomes using a reference genome resulted in as the authors stated "high quality genomes" with a high completeness.
Genome comparisons led to the identification of several phylogenomic clusters revealing one cluster that was presumably mainly responsible for the outbreak, however multiple sources of infection were not excluded. PCA analyses did not reveal correlations.
Response: Thank you. We’re very grateful for the positive feedback.
Point 2: As the authors discuss, the study is limited by the study design/methodology which did not allow to reconstruct the complete chromosomal/plasmidic content of the isolated strains. This is a severe limitation since genes for ABR or virulence very often reside on plasmids that can be transferred by conjugation. Additionally sequencing producing long reads (such as Oxford Nanopore) would allow de novo assembly of genomes including assigning chromosomal and plasmidic origins of genes contributing to virulence and/or antibiotic resistance. This should be clearly stated both in the introduction and discussion sections.
Response: We would like to thank the reviewer for their helpful comments and suggestions. As the reviewer references, while we use de novo assembly methods to generate a genome assembly and then a reference-based scaffolding method to order and finalize the genome sequence, we did not target plasmids either in library construction and/or DNA assembly methods. There are more specific ways to do that and we chose a more general approach to simply target generating an overall “complete” genome. It should be noted, however, that for all AMR/virulence gene investigations we included all scaffolds and any unscaffolded, but assembled contigs in that analysis. Regardless, we have edited the manuscript where we thought it best to address the reviewers concerns and to be clearer in our statements on methods used and study limitations.
Reviewer 3 Report
The paper presents a significant contribution to the field by offering a practical approach to identifying S.marcescnes sources in a correctional facility. The authors effectively demonstrate the application of their methodology and propose its implementation in correctional facilities nationwide, making it a valuable resource for the community.
The detection of S.marcescnes absence in potable water is commendable, as its presence could have led to severe consequences and widespread infections. However, I recommend the inclusion of controls to assess the efficiency of isolation and detection sensitivity. To establish the sensitivity range of the assay, conducting experiments with different CFUs of S.marcescnes spiked into water devoid of the organism and employing the same isolation and culturing method is essential.
Identifying S.marcescnes in improperly diluted CB64 is a crucial finding, offering practical insights for modifying procedures in correctional facility management. The authors' thorough analysis using whole genome sequencing, comparative genomics, and phylogenomics is commendable. Identifying clustered strains from clinically relevant samples highlights potential social conditions contributing to the agent's spread. Implementing the guidelines based on these findings could lead to effective measures for reducing the transmission of S.marcescnes in such environments.
Nevertheless, I propose an independent approach to corroborate the phylogenetic findings presented in the paper. Utilizing de-novo assemblies and in-silico MLST instead of reference-based assembly can offer an alternative perspective, as it is crucial to verify the accuracy of phylogroup clustering. This approach allows for a better understanding of potential differences or errors in the existing methodology. Furthermore, exploring non-mapped reads in reference-based assembly could provide valuable information on new or modified plasmids carrying virulence factors through horizontal gene transfer.
Overall, the authors' work is thorough and significant, providing valuable insights into identifying and characterizing S.marcescnes in a correctional facility. Addressing the suggested points would enhance the paper's credibility and further contribute to the scientific community's understanding of the subject matter. An explanation of the proposed points would be greatly appreciated and strengthen the overall impact of the study.
Author Response
Response to Reviewer 3 Comments
Point 1: The paper presents a significant contribution to the field by offering a practical approach to identifying S.marcescnes sources in a correctional facility. The authors effectively demonstrate the application of their methodology and propose its implementation in correctional facilities nationwide, making it a valuable resource for the community.
Response: Thank you! We’re very pleased to hear this. Providing a valuable resource for S. marcescens outbreak investigators was the main motivation for writing this paper.
Point 2: The detection of S.marcescnes absence in potable water is commendable, as its presence could have led to severe consequences and widespread infections. However, I recommend the inclusion of controls to assess the efficiency of isolation and detection sensitivity. To establish the sensitivity range of the assay, conducting experiments with different CFUs of S.marcescnes spiked into water devoid of the organism and employing the same isolation and culturing method is essential.
Response: Thank you for acknowledging our efforts to detect S. marcescens in potable water, particularly since there is no standard method. We typically analyze potable water in our laboratory using membrane filtration and culture media with known recovery efficiencies for coliforms and had not considered evaluating S. marcescens specifically. We did compare S. marcescens counts among diluted liquid samples that were comparable based on varied dilutions; however, your suggestion to assess detection and recovery efficiencies is one that we will keep in mind for future analyses.
Point 3: Identifying S.marcescens in improperly diluted CB64 is a crucial finding, offering practical insights for modifying procedures in correctional facility management. The authors' thorough analysis using whole genome sequencing, comparative genomics, and phylogenomics is commendable. Identifying clustered strains from clinically relevant samples highlights potential social conditions contributing to the agent's spread. Implementing the guidelines based on these findings could lead to effective measures for reducing the transmission of S.marcescens in such environments.
Response: Thank you! Leadership at California Correctional Care Services is educating correctional facilities statewide with information on proper use of CB64.
Point 4: Nevertheless, I propose an independent approach to corroborate the phylogenetic findings presented in the paper. Utilizing de-novo assemblies and in-silico MLST instead of reference-based assembly can offer an alternative perspective, as it is crucial to verify the accuracy of phylogroup clustering. This approach allows for a better understanding of potential differences or errors in the existing methodology. Furthermore, exploring non-mapped reads in reference-based assembly could provide valuable information on new or modified plasmids carrying virulence factors through horizontal gene transfer.
Response: For the scope of this study, we do not feel like it is necessary to conduct additional MLST analysis or plasmid investigation for the following reasons:
1) We have already verified phylogroup clustering bY corroborating the RealPhy tree with another SNP-based tree Maximum Likelihood tree and clustering with ANI comparison of the largest scaffolds from each assembly (all of these efforts showed consistent results with regard to clustering of phylogroups)/
2) While we recognize the value of exploring either an in-silico MLST or whole-genome MLST in generating types that could be used for association with other studies, we do not need such comparison for our work and we consider the use of our whole genome sequence as more data rich and detailed for the patterns of variations we are observing (in some cases as few as 2 SNPs) than a subsampled approach that is MLST-based. Our data is publicly available, however, so anyone else interested in generating an in silico or wg MLST for our samples will be able to.
3) We feel there is some confusion over the assembly process and have tried to edit our language where appropriate, but, to be clear, we conducted a de novo assembly and then used a published genome to scaffold that assembly. In our final assembly, we did not discard any unmapped or unordered contigs. Thus, while we didn’t target a specific plasmid assembly method, if there was enough sequence in our data it still could have been assembled and was included in our AMR/virulence analysis. Thus, we feel from a comparative perspective, our results should not be overly biased by this approach. That said, a different effort to target such structures could be warranted but is beyond the scope of a minor revision as it could involve significant effort in both new data acquisition and analysis methods to properly investigate.
Point 5: Overall, the authors' work is thorough and significant, providing valuable insights into identifying and characterizing S.marcescnes in a correctional facility. Addressing the suggested points would enhance the paper's credibility and further contribute to the scientific community's understanding of the subject matter. An explanation of the proposed points would be greatly appreciated and strengthen the overall impact of the study.
Response: We appreciate the reviewers comments and insights and have edited the text in the discussion to better recognize and identify the limitations of our current approach. Thank you for taking time to do a thorough and constructive review.